# Do Where The Elderly Live Matter? Factors Associated with Diet Quality among Korean Elderly Population Living in Urban Versus Rural Areas

**DOI:** 10.3390/nu12051314

**Published:** 2020-05-05

**Authors:** Sohyun Park, Hyun Ja Kim, Kirang Kim

**Affiliations:** 1Department of Food Science and Nutrition, Hallym University, 24252 Chuncheon, Korea; sopark@hallym.ac.kr; 2Department of Food and Nutrition, Gangneung-Wonju National University, 25457 Gangneung, Korea; 3Department of Food Science and Nutrition, Dankook University, 31116 Cheonan, Korea

**Keywords:** diet quality, healthy eating index, urban, rural, older adults, personal factors

## Abstract

This study aimed to examine whether there is an area difference on diet quality among the Korean elderly population. The effect of personal factors on diet quality is also estimated and compared between rural and urban areas. A cross-sectional data from the 2013–2015 Korea National Health and Nutrition Examination Survey (KNHANES) was used for this study. The participants were older adults aged ≥ 65 years (*n* = 3207) who participated in the KNHANES. Urban and rural areas classified the region and the Korean Healthy Eating Index (KHEI) assessed the diet quality. Personal factors that were related to diet quality included socio-demographic factors, health behaviors, and health conditions. This study found that the diet quality was different between urban and rural areas in the Korean elderly population, showing a higher mean of KHEI scores in urban areas than rural areas (67.3 for urban seniors, 63.6 for rural seniors, *p* < 0.001), and the regional difference was still significant, even after adjusting for the personal factors (*p* < 0.001). Different sets of personal factors were found to be significant that explain the diet quality of participants between areas, such as economic resources, walking exercise, and perceived oral health status in urban areas, and age and food insecurity in rural areas. In conclusions, this study found that there was a regional disparity in diet quality and some personal factors affecting diet quality were dependent on areas, which implied that regional environment with diverse contexts could influence diet quality. These findings emphasize the need to provide targeted intervention programs that take into account both the characteristics of individuals and local food environments in order to improve the overall diet quality in older adults.

## 1. Introduction 

The intake of various foods and nutrients and the balanced quality of meals are important in preventing chronic diseases. According to the recent reports from the Global Burden of Disease Study in 2017, poor diet accounted for more than 50% of deaths and 66% of disability-adjusted life years [1]. A recent Korean study also confirmed that poor diet quality was associated with a higher mortality for all-cause, cardiovascular disease, and cancer, while using 12-year follow-up mortality data from a nationally representative sample [2]. 

When considering the importance of healthy diet on public health, various methods for assessing the overall quality of diet have been developed [3]. Among them, the healthy eating index is one of the valid tools to assess varied and balanced dietary intakes, which could be used to evaluate dietary programs and policies [4,5,6,7]. Recently, the Korean Healthy Eating Index (KHEI) was developed [8] and it is currently used in the Korea National Health and Nutrition Examination Survey (KNHANES) [9] to assess diet quality in Korean populations.

In the elderly, dietary intake would be adversely affected by the physiological, economic, and societal factors with aging [10]. Nutrition for older adults becomes a particularly essential component for the good maintenance of health and functional capacity [11]. Generally, factors affecting diet quality among older adults have been known as personal factors, such as socio-demographic factors, life style, and psychosocial factors, and so on [12,13,14]. However, previous studies reported that personal factors influencing diet quality was different by regions [15,16,17,18], which implies the interaction effect of personal characteristics and living environments.

The health status has generally improved in Korea, but there are significant differences in health outcomes between regional areas, showing better health in urban residents than rural residents, which result in increased problems of regional health inequality [19,20]. Several studies reported that interactions with personal characteristics and various social environments within areas that are related to individual health, rather than the geographical location itself, could be explain the difference [19,20,21]. Identifying factors contributing to regional dietary inequality would help to improve population health status and reduce the inequality.

Since healthy dietary intake is an important and readily modifiable determinant on health outcomes, understanding the contributing factors for regional dietary disparity could provide insight for planning intervention strategies, which could ultimately address the health and nutrition inequality. However, there is very little research on the regional dietary disparity in the Korean elderly population [16,22], and even no study on the personal factors influencing the regional dietary disparity. Therefore, this study aimed to investigate whether there is an area difference on diet quality and the effect of personal factors on diet quality is different by areas among the Korean elderly population, while using the KNHANES data.

## 2. Methods and Materials

### 2.1. Study Design and Subjects

This study was conducted using data from the KNHANES IV (2013–2015), a nationwide cross-sectional survey conducted by the Korea Centers for Disease Control and Prevention (KCDC). This survey was established to monitor health and nutritional status of the non-institutionalized Korean population since 1998. Using a stratified and multi-stage clustered probability sampling method, a total 4509 elderly participants aged ≥ 65 years have participated in KNHANES from 2013 to 2015. Among the elderly adults, we excluded those who reported a diagnosis of any cancer, cardiovascular diseases, renal failure, or liver cirrhosis (*n* = 863), and who had an implausible daily caloric intake (<500 kcal/day or >5000 kcal/day, *n* = 261). Among the remaining 3385 older adults, we excluded 178 who missed health examination or nutrition survey. Finally, a total of 3207 elderly adults (1339 men and 1868 women) were included in the current analysis.

A written informed consent regarding the survey was obtained from all participants. As the KCDC institutional review board (IRB) approved all procedures and protocols for the survey (2013-07CON-03-4C, 2013-12EXP-03-5C, 2015-01-02-6C), the additional IRB process for this study has not been required.

### 2.2. Methods and Variables

#### 2.2.1. Socio-demographic Factors, Health Behaviors and Health Conditions 

In this study, socio-demographic information related to diet quality was divided, as follows: age (65–74 or ≥75 years), sex (men or women), household types (single or non-single), marital status (married/living as married, separated/divorced, or widowed), education level (≤elementary school, middle school, high school, or ≥college), household income (quartiles), job status (non-manual job, manual job, or unemployment, including housewives and students), home ownership (yes or no), beneficiaries of national basic livelihood (yes or no), and food assistance program participation (yes or no). The area of residence was classified into urban (cities divided into *dong*) or rural (towns called *eup* or townships called *myeon*) by their administrative districts. 

We considered several health behavioral factors (smoking, alcohol drinking, walking exercise), experience of nutritional education or counseling, eating with others, and health conditions (limited social activity due to disability, obesity, chronic disease, depression, and poor oral health) because health behaviors and health conditions have been reported to have association with diet quality [7,23,24]. Smoking was classified into three categories (never, past, or current) and alcohol drinking was classified into two categories (yes, including past, or no). Walking exercise was classified as yes (more than 10 min each time, more than 30 min each day, and five days or more per week during the past one week) or no. The experience of nutrition education or counseling during the last year was classified into two categories (yes or no). Eating with others was classified into two categories (yes or no) using a question of “did you usually have dinner with others in the last year?”. Limited social activity due to disability was surveyed using a question of “are you restricted in your daily activities and social activities due to health problems or physical or mental disabilities?” and was classified into two categories (yes or no). Body mass index (BMI) was calculated as the ratio of weight to height squared (kg/m^2^) and it was classified into four categories (underweight of <18.5 kg/m^2^, normal of 18.5–22.9 kg/m^2^, overweight of 23.0–24.9 kg/m^2^, or obesity of ≥25 kg/m^2^), according to the re-defined criteria of the World Health Organization for obesity in the Asia-Pacific region [25]. The participants was defined as a chronic disease group if they have at least one of the following disease: hypertension (defined as an average SBP and/or DBP ≥ 140/90 mmHg or the presence of anti-hypertensive agents), diabetes (defined as fasting plasma glucose of ≥126 mg/dL, a previous diagnosis of diabetes by physician, or current use of anti-diabetic agents or insulin), or hypercholesterolemia (defined as plasma total cholesterol of ≥240 mg/dL or current use of cholesterol-lowering agents). Depression and perceived poor oral health were surveyed by self-administered questionnaire. If the participants answered “yes” on a question of “have you experienced a continuous feeling of sadness or despair for over two weeks that interfered with your daily life in the last year?”, they were classified as a depression group. The participants were classified as a perceived poor oral health group if they selected “very uncomfortable” or “uncomfortable” of five scales (“very uncomfortable”, “uncomfortable”, “neither”, “comfortable”, and “very comfortable”) on a question of “do you feel uncomfortable about chewing food due to the problems with your mouth, teeth, dentures, and gums?”. 

It has been reported that household food insecurity was associated with low diet quality [26], so we consider food insecurity as a related factor in this study. Using an 18-items questionnaire on household food security, individuals within household were classified as a food-secure group if the total score was 0–2 and a food-insecure group if the total score was 3 or more. 

#### 2.2.2. Korean Healthy Eating Index (KHEI)

The KHEI was developed to evaluate diet quality by scoring adherence to dietary guidelines for Korean by the KCDC [8]. The KHEI is composed of total 14 components, which includes eight adequacy components for recommended foods and nutrients, three moderation components for restricted food and nutrient, and three balance components for energy intake. The KHEI total score is calculated to be 0–100 points. Among eight adequacy components, three components are given 0–10 points (‘have breakfast’, ‘meat, fish, eggs, and beans intake’, and ‘milk and milk products intake’), and 0–5 points for five components related to grains, fruits, and vegetables (‘mixed grains intake’, ‘total fruit intake’, ‘fresh fruit intake’, ‘total vegetable intake’, ‘vegetables intake excluding Kimchi and pickled vegetables’). All three moderation components (‘% of energy from saturated fatty acid’, ‘sodium intake’, ‘% of energy from sweets and beverages’) are given 0–10 points and all of three balance components (‘% of energy from carbohydrate’, ‘% of energy from fat’, ‘energy intake’) are given 0–5 points.

### 2.3. Statistical Analysis

All data from the 2013–2015 KNHANES were pooled into one dataset and then sampling weights were calculated by dividing the sampling weight assigned to participant by the number of survey years. The new sampling weights were applied to all analyses. P-values for % differences were calculated by chi-square test. All the means and standard error (SE) of KHEI score were presented as age- and/or sex-adjusted values and p-values for mean differences of KHEI score were calculated by multivariate linear regression after adjusting for age and/or sex. Tukey’s test was used for multiple comparisons of mean differences. For identifying the effect of region (urban and rural area) on KHEI score, regression coefficients (β) and p-values were calculated by multivariate linear regression analysis after adjusting for confounding factors. Significant variables affecting to KHEI score by area were selected by stepwise linear regression analysis. All of the analyses were conducted using the IBM SPSS Statistics version 23 (IBM SPSS INC, Armonk, NY, USA) to analyze data from complex sample survey. Two-sided *p*-values < 0.05 were considered to be statistically significant.

## 3. Results

### 3.1. General Characteristics between Urban and Rural Participants

The general characteristics between urban and rural participants are shown in Table 1. The proportion of residents in each area was 71.8% for urban area and 28.2% for rural area. In terms of socio-demographic factors, the participants in urban areas had higher education level (*p* < 0.001) and household income (*p* < 0.001), and more participated in food assistance programs (*p* = 0.001), but the participants in rural areas were older (*p* < 0.001) and had higher proportion of living alone (*p* = 0.046), widowed (*p* = 0.019), working manual job (*p* < 0.001), and owning home (*p* < 0.001). For health behaviors and health conditions, rural participants had lower walking exercise (*p* < 0.001) and higher limited social activity due to disability (*p* = 0.005) than urban participants, but higher proportion of eating with others (*p* = 0.001) and lower prevalence of chronic disease (*p* = 0.040).

### 3.2. The KHEI Scores by Sex, Age Group, and Areas

Table 2 presents the comparison of mean score of KHEI between urban and rural areas. The total mean score of KHEI was 66.4 (SE = 0.29) and it was different between urban and rural areas, showing a higher score in urban participants (67.3 ± 0.33 for urban area and 63.6 ± 0.50 for rural area, *p* < 0.001). The higher score in urban area was consistent in both the sex and age groups. The score of each KHEI item was also generally higher in urban participants, except for item of having breakfast, as compared with rural participants. The regional difference for KHEI was more apparent in women than men, regarding the intake of vegetables, excluding Kimchi and pickled vegetables, meat fish eggs beans, and milk and dairy products. For age groups, there was a regional difference, specifically in total and fresh fruit intake for participants aged 75 years or more, and in milk and dairy products intake for those aged 65–74 years. 

### 3.3. The KHEI Scores and Their Related Factors by Areas

Table 3 shows that most of the socio-demographic factors, health behaviors, and health conditions were associated with KHEI total scores. Specifically, the participants who are married or living as married, have longer education, have higher household income, have non-manual job, have home ownership, have food security, and do not participant in the national basic livelihood and food assistance programs show higher total HEI scores. As for health behaviors and health conditions, the participants who never smoked, perform walking exercise, eat with others, do not have limited social activity due to disability, are not underweight, are not depressed, and have a healthy oral condition show higher total HEI scores. 

When multiple comparisons were performed within each area, urban participants showed similar patterns of significant differences by various socio-demographic, health behaviors, and health conditions, as shown among all participants. Among rural participants, differences in KHEI by these various factors were not as salient as urban participants. For example, unlike urban participants, rural participants with different status in home ownership, participation in the national basic livelihood and food assistance programs, smoking status, and weight status did not have statistically significant differences in KHEI scores.

### 3.4. The Regional Effect of KHEI 

Table 4 shows the regional effect on KHEI after adjusting for all the potential confounders, as presented in Table 3. After adjusting for key socio-demographic factors, health behaviors, and health conditions, participants from rural areas had lower HEI scores than participants from urban areas by 2.6 points. When stratified by sex, the difference between rural and urban was 2.5 points among men and 2.7 points among women. When stratified by age group, the difference was 2.5 points among 65–74 years old and 3.0 points among the participants that were aged 75 years or older.

### 3.5. Factors that Affect KHEI Scores in Urban and Rural Areas

Table 5 illustrates the explanatory factors that are associated with the KHEI total scores in urban and rural areas, while using a stepwise regression. More factors were selected in urban areas than in rural areas. Sex and education level were the common factors, selected in both areas. In addition to these factors, marital status, household income, home ownership, walking exercise, and perceived oral health were found to be associated with the KHEI scores in urban participants. Age and food security status were the additional related factors for KHEI, selected in rural participants. 

In urban areas, being women, having higher education, being married or cohabiting with a partner, having higher income, having home ownership, walking regularly, and having healthy oral condition were the factors that explain higher diet quality (adjusted R^2^ = 0.15). In rural areas, being younger, being women, having higher education, and having food security were the factors that explain the higher diet quality (adjusted R^2^ = 0.13). 

## 4. Discussion

Understanding factors contributing to regional dietary inequality could be important in reducing the inequality because of increasing health and dietary disparity. The objective of this study was to examine whether there is a regional difference on diet quality. In addition, the effect of personal factors on diet quality was compared between the elderly in urban and rural areas. This study found that the diet quality was different between urban and rural areas in the Korean elderly population, showing higher diet quality in urban areas than rural areas. The regional differences were still significant, even after adjusting for various personal factors. Different sets of personal factors were found to be significant in rural and urban areas that explain the diet quality of participants.

It is noteworthy that the main differences in individual items for KHEI scores between urban and rural older adults mostly occur in adequacy items and the percent of energy intake from carbohydrate and fat. No statistically significant differences were found in moderation items between two regions. In adequacy items, specifically, intakes of total fruits, vegetables, excluding Kimchi and other pickled vegetables, protein sources, and milk and dairy products were significantly lower among rural older adults. These food items are key in assuring the consumption of essential micronutrients, fiber, and protein, which are critical in preventing aging-related health conditions including sarcopenia [27], frailty [28], and cardiovascular diseases [29,30]. 

Our study showing the higher diet quality in urban older adults, especially for the adequacy of various foods among KHEI items was consistent with other previous studies [22,31,32,33]. The poor food access in rural older adults could contribute to lower intakes of fruit, milk, and dairy products when compared to urban older adults [15,34,35,36]. Economic disparity can play a role in food availability and accessibility in some regions [21,37,38]. In addition, mainly relying on foods from farming or home gardening in rural areas might result in limited variety of foods [39]. Individual efforts may not be enough in increasing the consumption of these food items, such as protein sources, such as meat, fish, eggs, beans, and milk and dairy products, due to these various environmental and economic constraints. Therefore, it would be necessary for government agencies or charity-based organizations to incorporate these food items in their food assistance programs for vulnerable elderly populations in rural areas. 

It would be meaningful to examine whether the current food assistant programs in Korea meet the actual need of target population in terms of diet quality. Although the study subjects were not the elderly, previous study using a qualitative method that was conducted in Korea revealed that the beneficiaries of government food assistant programs still perceived the lack of certain food items in their diet despite of the support program. These items were mostly protein sources, such as meat, poultry, and dairy products [40]. Therefore, the food items provided through the current program for the elderly population, especially in rural areas, should also be evaluated if the program actually meets the needs of participants and contributes to their diet quality.

As mentioned earlier, the results showed that the dietary quality was lower among the rural participants than the urban ones. Specifically, the consumption of protein and vegetables was lower among the rural elderly populations. However, the prevalence of chronic diseases was lower in rural areas. This result might be explained by the inherent nature of cross-sectional study design that cannot assure the causality. It is also hypothesized that the less available medical services in rural areas may hinder the early detection and management of chronic diseases among rural participants. The prevalence of chronic diseases in this study was measured with a set of questions answered by the participants, not assessed by medical professionals. 

Different personal factors were found to be significant in explaining diet quality in urban and rural areas. In urban areas, the economic resources, such as household income and home ownership, were significantly associated with the total KHEI scores. This might be due to relatively higher living and housing cost in urban areas and, consequently, fewer budgets on food purchase, which further sacrifice diet quality in urban regions [41]. One study that was conducted in the States showed that poor families sacrifice their caloric intake during cold weather due to high expenses on heating [42]. Similar situations may also be occurring in poor urban households in Korea. Therefore, additional supporting programs, such as housing or heating assistance programs, to stretch out budget that can be spent on food might be an effective strategy for increasing diet quality among urban elderly populations.

On the other hand, the diet quality of rural elderly participants was not explained by these economic factors in this study. A high percentage of rural participants who are engaging in farming or gardening might weaken the effect of economic difficulties on diet quality [43]. The culture of sharing foods among rural elderly participants might also explain this difference in economic effect on diet quality [15,44]. 

Other personal factors influencing diet quality differently between the regions were walking exercise and perceived oral health. These factors only had a significant effect on diet quality in urban older adults. One of the possible explanations would be that urban seniors are more interested in health promoting practices than rural seniors [45,46] and, thus, urban seniors who practice healthier behaviors might want to pursue healthier eating. Further research is needed to examine the reasons. However, this finding could suggest that targeting other health behaviors, such as walking exercise or behaviors related to oral health, coupled with healthy eating intervention, would synergistically improve diet quality among urban elderly populations. However, it should be noted that the prevalence of walking exercise was significantly lower among rural participants. The walking exercise does not include the manual labor associated with farming in rural areas. Therefore, it would be interesting to estimate the association between total physical activities and dietary quality and evaluate the differences between the type of physical activities and its relations with dietary practices. 

Regional dietary disparity would be explained by the personal factors living in the areas. However, there are still significant effects of areas on diet quality after adjusting for personal factors, as shown in this study. This reason might be explained by community environment related to food accessibility. Several previous studies reported that the availability and accessibility of grocery stores to purchase healthy and safe foods in the neighborhood, and the price of the foods, were significantly associated to healthy food intakes [34,36,47,48]. Interestingly, the effect of community food environment on healthy diets was stronger in rural areas than urban areas [49]. In this study, food insecurity was an important risk factor of diet quality in only rural older adults. A previous study found that limited food accessibility due to long distance to grocery stores was a significant risk factor of food insecurity in the Korean rural elderly population [43]. Therefore, the food insecurity in rural older adults could be attributed not only by their low socioeconomic status, but also by poor local food environment, which can ultimately contribute to lower diet quality. This result underscores the need for tailored interventions that consider food environments, especially for rural older adults suffering from food insecurity. 

Several limitations need to be considered when interpreting the findings of the present study. First, its cross-sectional design limits the causal interpretation of relationship between diet quality and related factors. Second, other plausible factors that could be related to diet quality, such as regional food environment and social network, were not included in the analysis, due to the lack of information. In this point of view, the relationship of food and social environments with diet quality should be addressed in future studies.

In conclusion, this study found that there was regional dietary disparity and some personal factors affecting diet quality were dependent on areas, implying that regional environments with diverse contexts could influence diet quality. These findings emphasize the need to provide targeted intervention programs that take into account both the characteristics of individuals and local food environments in order to improve overall diet quality in older adults. In particular, rural areas have a higher proportion of older people, which lead to poorer health and nutritional status than other areas, so that more attention should be paid to nutritional policy in these vulnerable areas to reduce the nutritional inequality.

## Figures and Tables

**Table 1 nutrients-12-01314-t001:** General characteristics of adults aged 65 years and older, 2013–2015 Korea National Health and Nutrition Examination Survey (KNHANES).

Variables	Total (*n* = 3207)		Urban (*n* = 2302)		Rural (*n* = 905)		*p*-Value ^2^
**Sex**							
Men	43.3	(1.0) ^1^	43.3	(1.1)	43.3	(1.8)	0.977
Women	56.7	(1.0)	56.7	(1.1)	56.7	(1.8)	
**Age group**							
65–74 yrs	63.3	(1.0)	66.1	(1.2)	55.1	(1.9)	<0.001
≥75 yrs	36.7	(1.0)	33.9	(1.2)	44.9	(1.9)	
Household types							
Single	17.6	(0.7)	16.8	(0.8)	20.1	(1.5)	0.046
Non-single	82.4	(0.7)	83.2	(0.8)	79.9	(1.5)	
Marital status							
Married/living as married	64.5	(1.1)	64.3	(1.3)	65.0	(2.1)	0.019
Separated/Divorced	3.3	(0.4)	3.9	(0.4)	1.6	(0.4)	
Widowed	32.2	(1.1)	31.8	(1.3)	33.4	(2.1)	
Education level							
≤Elementary school	61.9	(1.2)	58.2	(1.5)	73.9	(2.3)	<0.001
Middle school	13.2	(0.8)	14.0	(1.0)	10.6	(1.2)	
High school	16.7	(0.9)	18.2	(1.1)	11.9	(1.5)	
≥College	8.2	(0.7)	9.6	(0.8)	3.7	(1.0)	
Household income							
Q1 (lowest)	24.3	(1.0)	21.9	(1.1)	31.8	(2.2)	<0.001
Q2	24.6	(1.0)	23.6	(1.2)	27.7	(1.9)	
Q3	24.6	(1.0)	25.9	(1.1)	20.7	(1.8)	
Q4 (highest)	26.4	(1.2)	28.6	(1.4)	19.8	(2.0)	
Job status							
Non-manual job	3.2	(0.4)	3.7	(0.5)	1.6	(0.5)	<0.001
Manual job	29.2	(1.2)	24.2	(1.2)	45.6	(2.9)	
Unemployment ^3^	67.6	(1.2)	72.1	(1.2)	52.8	(2.9)	
Home ownership							
Yes	73.9	(1.3)	71.2	(1.6)	82.2	(1.7)	<0.001
No	26.1	(1.3)	28.8	(1.6)	17.8	(1.7)	
Beneficiaries of national basic livelihood							
Yes	12.6	(1.0)	12.3	(1.2)	13.8	(2.0)	0.514
No	87.4	(1.0)	87.7	(1.2)	86.2	(2.0)	
Food assistance program participation							
Yes	6.7	(0.6)	7.6	(0.7)	4.0	(0.7)	0.001
No	93.3	(0.6)	92.4	(0.7)	96.0	(0.7)	
Food security							
Yes	89.5	(0.7)	89.8	(0.9)	88.8	(1.3)	0.528
No	10.5	(0.7)	10.2	(0.9)	11.2	(1.3)	
Smoking							
Never	61.5	(1.0)	61.6	(1.2)	61.0	(1.8)	0.939
Past	27.0	(0.9)	26.8	(1.1)	27.5	(1.6)	
Current	11.6	(0.8)	11.6	(0.9)	11.5	(1.3)	
Alcohol drinking							
Yes	73.3	(0.9)	73.8	(1.1)	71.8	(1.9)	0.364
No	26.7	(0.9)	26.2	(1.1)	28.2	(1.9)	
Walking exercise							
Yes	39.2	(1.1)	42.6	(1.3)	28.3	(1.8)	<0.001
No	60.8	(1.1)	57.4	(1.3)	71.7	(1.8)	
Nutritional education							
Yes	4.4	(0.4)	4.7	(0.5)	3.3	(0.6)	0.094
No	95.6	(0.4)	95.3	(0.5)	96.7	(0.6)	
Eating with others							
Yes	67.6	(1.1)	65.8	(1.3)	72.8	(1.7)	0.001
No	32.4	(1.1)	34.2	(1.3)	27.2	(1.7)	
Limited social activity due to disability							
Yes	14.9	(0.8)	13.5	(0.9)	19.3	(2.0)	0.005
No	85.1	(0.8)	86.5	(0.9)	80.7	(2.0)	
Weight status, BMI							
Underweight	3.5	(0.4)	3.1	(0.4)	4.5	(0.9)	0.124
Normal	35.3	(1.1)	34.6	(1.2)	37.4	(2.1)	
Overweight	27.3	(0.9)	28.1	(1.1)	24.7	(1.5)	
Obesity	34.0	(0.9)	34.2	(1.1)	33.3	(1.7)	
Chronic disease							
Yes	72.7	(1.1)	73.9	(1.3)	68.6	(2.3)	0.040
No	27.3	(1.1)	26.1	(1.3)	31.4	(2.3)	
Depression							
Yes	14.4	(1.0)	13.6	(1.1)	17.0	(2.3)	0.160
No	85.6	(1.0)	86.4	(1.1)	83.0	(2.3)	
Perceived poor oral health							
Yes	44.2	(1.2)	43.1	(1.4)	47.8	(2.2)	0.071
No	55.8	(1.2)	56.9	(1.4)	52.2	(2.2)	

KNHANES, Korea National Health and Nutrition Examination Survey. ^1^ % (SE), which were calculated by applying sampling weights assigned to individual participants in the nutrition survey and health examination. ^2^
*p*-values for % differences between urban and rural area were calculated using the chi-square test. ^3^ Including housewives or students.

**Table 2 nutrients-12-01314-t002:** Mean value of each Korean Healthy Eating Index (KHEI) item by sex and region.

Variables	Total Area	Total		*p*-Value ^2^	Men		*p*-Value ^2^	Women		*p*-Value ^2^	65–74 yrs		*p*-Value ^2^	≥75 yrs		*p*-Value ^2^
		Urban	Rural		Urban	Rural		Urban	Rural		Urban	Rural		Urban	Rural	
No. of subjects	3207	2302	905		960	379		1342	526		1481	498		821	407	
Total score	66.4(0.29)	67.3(0.33) ^1^	63.6(0.50)	<0.001	66.2(0.45)	62.9(0.60)	<0.001	68.1(0.42)	64.2(0.65)	<0.001	66.4(0.68)	63.2(0.82)	<0.001	62.9(0.80)	59.6(0.97)	<0.001
Adequacy item																
Have breakfast	9.5(0.05)	9.4(0.06)	9.7(0.06)	0.003	9.6(0.07)	9.7(0.07)	0.222	9.3(0.09)	9.6(0.08)	0.002	9.7(0.11)	9.9(0.13)	0.012	9.8(0.11)	10.2(0.13)	0.001
Mixed grains intake	2.8(0.06)	3.0(0.06)	2.2(0.11)	<0.001	3.1(0.08)	2.2(0.14)	<0.001	2.9(0.07)	2.2(0.12)	<0.001	3.1(0.12)	2.5(0.16)	<0.001	2.8(0.16)	2.0(0.18)	<0.001
Total fruit intake	2.6(0.06)	2.6(0.07)	2.4(0.10)	0.015	2.3(0.09)	2.0(0.13)	0.061	2.9(0.08)	2.6(0.13)	0.056	2.1(0.13)	1.9(0.16)	0.105	1.8(0.16)	1.5(0.18)	0.015
Fresh fruit intake	2.7(0.06)	2.8(0.07)	2.5(0.11)	0.102	2.5(0.10)	2.3(0.14)	0.162	2.9(0.08)	2.8(0.13)	0.224	2.3(0.14)	2.2(0.17)	0.671	2.3(0.18)	1.9(0.20)	0.026
Total vegetable intake	3.7(0.03)	3.7(0.04)	3.7(0.06)	0.626	3.8(0.06)	3.7(0.08)	0.699	3.7(0.05)	3.7(0.08)	0.717	3.9(0.09)	3.9(0.10)	0.774	3.5(0.11)	3.5(0.13)	0.833
Vegetables intake excluding Kimchi and pickled vegetables	3.4(0.04)	3.5(0.04)	3.2(0.08)	0.001	3.3(0.06)	3.1(0.10)	0.078	3.6(0.05)	3.3(0.10)	0.001	3.3(0.09)	3.1(0.12)	0.022	2.8(0.13)	2.6(0.15)	0.013
Meat, fish, eggs and beans intake	6.4(0.08)	6.5(0.10)	6.0(0.15)	0.005	6.6(0.13)	6.3(0.18)	0.199	6.5(0.12)	5.8(0.19)	0.003	7.3(0.19)	7.0(0.24)	0.078	5.6(0.24)	5.2(0.30)	0.093
Milk and milk products intake	2.4(0.09)	2.5(0.10)	1.9(0.17)	0.002	2.2(0.14)	1.8(0.24)	0.128	2.8(0.13)	2.0(0.18)	0.001	2.1(0.22)	1.4(0.25)	<0.001	1.7(0.25)	1.2(0.35)	0.080
Moderation item																
% of energy from saturated fatty acid	9.4(0.04)	9.5(0.05)	9.4(0.08)	0.871	9.4(0.07)	9.3(0.12)	0.628	9.5(0.06)	9.5(0.10)	0.828	9.1(0.12)	9.2(0.14)	0.550	9.6(0.11)	9.6(0.15)	0.718
Sodium intake	7.2(0.08)	7.2(0.09)	7.2(0.15)	0.772	6.4(0.14)	6.3(0.20)	0.813	7.9(0.10)	7.9(0.17)	0.808	5.2(0.20)	5.1(0.22)	0.507	6.4(0.25)	6.5(0.26)	0.683
% of energy from sweets and beverages	9.5(0.04)	9.5(0.05)	9.4(0.07)	0.269	9.5(0.07)	9.3(0.11)	0.100	9.5(0.06)	9.5(0.09)	0.962	9.5(0.10)	9.4(0.13)	0.305	9.4(0.16)	9.4(0.16)	0.662
Balance of energy intake item																
% of energy from carbohydrate	1.4(0.04)	1.5(0.05)	1.1(0.07)	<0.001	1.8(0.08)	1.5(0.10)	0.022	1.3(0.06)	0.9(0.08)	<0.001	2.2(0.12)	1.9(0.13)	0.005	1.5(0.13)	1.1(0.13)	<0.001
% of energy from fat	2.2(0.05)	2.4(0.06)	1.8(0.09)	<0.001	2.6(0.09)	2.2(0.13)	0.009	2.2(0.07)	1.5(0.10)	<0.001	3.0(0.13)	2.5(0.16)	<0.001	2.3(0.16)	1.8(0.17)	<0.001
Energy intake	3.1(0.05)	3.2(0.06)	3.1(0.08)	0.439	3.3(0.08)	3.2(0.11)	0.669	3.1(0.07)	3.0(0.11)	0.491	3.4(0.13)	3.3(0.14)	0.378	3.2(0.16)	3.2(0.20)	0.880

KHEI, Korean Healthy Eating Index. All means (SE) were calculated by applying sampling weights assigned to individual participants in the nutrition survey and health examination. ^1^ age- and/or sex-adjusted means (SE), excepting for total area values. ^2^
*p*-values for mean KHEI differences between urban and rural area were calculated using multivariate linear regression after adjusting for age (continuous) and sex. Sex and age were not adjusted in the analysis by sex and age group, respectively.

**Table 3 nutrients-12-01314-t003:** The mean value of KHEI total score by factors related to KHEI.

Variables	Total		*p*-Value ^2^	Urban		Rural		*p*-Value ^3^
	Mean	(SE) ^1^		Mean	(SE) ^1^	Mean	(SE) ^1^	
Household type								
Single	64.3	(0.6)	0.001	65.6	(0.8) ^ab^	60.9	(1.0) ^c^	<0.001
Non-single	66.6	(0.3)		67.5	(0.4) ^a^	64.1	(0.5) ^b^	
Marital status								
Married/living as married	67.3	(0.4) ^a^	<0.001	68.3	(0.4) ^a^	64.5	(0.5) ^b^	<0.001
Separated/Divorced	63.2	(1.3) ^b^		63.1	(1.5) ^bc^	65.0	(3.0) ^abc^	
Widowed	64.6	(0.5) ^b^		65.6	(0.6) ^b^	61.5	(0.8) ^c^	
Education level								
≤Elementary school	64.1	(0.4) ^a^	<0.001	65.0	(0.5) ^a^	62.1	(0.5) ^b^	<0.001
Middle school	67.9	(0.8) ^b^		68.5	(0.9) ^cd^	65.2	(1.3) ^abc^	
High school	70.9	(0.6) ^c^		71.6	(0.7) ^d^	67.6	(1.0) ^ac^	
≥College	72.6	(0.9) ^c^		72.7	(1.0) ^d^	71.2	(2.1) ^acd^	
Household income								
Q1 (lowest)	63.0	(0.5) ^a^	<0.001	63.8	(0.7) ^abc^	61.2	(0.8) ^b^	<0.001
Q2	65.6	(0.5) ^b^		66.4	(0.6) ^c^	63.3	(0.7) ^ab^	
Q3	67.0	(0.5) ^b^		67.7	(0.6) ^cde^	64.4	(0.9) ^abc^	
Q4 (highest)	69.4	(0.5) ^c^		70.0	(0.5) ^d^	66.4	(1.0) ^ac^	
Job status								
Non-manual job	73.1	(1.2) ^a^	<0.001	74.5	(1.2) ^a^	63.4	(3.2) ^bcd^	<0.001
Manual job	65.0	(0.4) ^b^		65.9	(0.6) ^cd^	63.5	(0.6) ^b^	
Unemployment ^4^	66.9	(0.4) ^c^		67.7	(0.4) ^d^	63.5	(0.7) ^bc^	
Home ownership								
Yes	67.1	(0.3)	<0.001	68.4	(0.4) ^a^	63.8	(0.5) ^b^	<0.001
No	63.7	(0.6)		64.1	(0.7) ^b^	61.8	(1.1) ^b^	
Beneficiaries of national basic livelihood								
Yes	63.3	(0.8)	<0.001	63.7	(1.0) ^a^	62.2	(1.1) ^a^	<0.001
No	66.7	(0.3)		67.7	(0.4) ^b^	63.7	(0.6) ^a^	
Food assistance program participation								
Yes	63.1	(1.0)	0.001	63.4	(1.1) ^a^	60.6	(1.3) ^a^	<0.001
No	66.5	(0.3)		67.5	(0.3) ^b^	63.6	(0.5) ^a^	
Food security								
Yes	66.8	(0.3)	<0.001	67.7	(0.3) ^a^	64.2	(0.5) ^b^	<0.001
No	62.1	(0.8)		63.2	(1.0) ^b^	59.0	(1.0) ^c^	
Smoking								
Never ^a^	67.8	(0.5) ^a^	<0.001	68.8	(0.5) ^a^	64.5	(0.7) ^bc^	<0.001
Ex ^b^	66.1	(0.6) ^a^		67.1	(0.7) ^ab^	63.2	(0.9) ^c^	
Current ^c^	62.1	(1.0) ^b^		62.2	(1.2) ^c^	61.8	(1.1) ^c^	
Alcohol drinking								
Yes	66.8	(0.3)	0.164	67.7	(0.4) ^a^	64.0	(0.5) ^bc^	<0.001
No	65.8	(0.6)		66.9	(0.7) ^ab^	62.8	(0.9) ^c^	
Walking exercise								
Yes	68.2	(0.4)	<0.001	69.2	(0.5) ^a^	63.1	(0.9) ^b^	<0.001
No	65.5	(0.4)		66.2	(0. 5)^c^	63.8	(0.6) ^b^	
Nutritional education and counseling								
Yes	68.2	(1.1)	0.058	68.9	(1.1) ^a^	64.9	(3.0) ^ab^	<0.001
No	66.2	(0.3)		67.1	(0.3) ^a^	63.5	(0.5) ^a^	
Eating with others								
Yes	66.9	(0.3)	<0.001	68.0	(0.4) ^a^	64.0	(0.5) ^bc^	<0.001
No	64.7	(0.4)		65.4	(0.5)^b^	62.0	(0.9)^c^	
Limited social activity due to disability								
Yes	64.7	(0.7)	0.006	65.3	(0.9) ^ab^	63.4	(0.9) ^a^	<0.001
No	66.8	(0.3)		67.8	(0.4) ^b^	63.6	(0.5) ^a^	
Weight status, BMI								
Underweight	62.2	(1.2) ^a^	0.002	62.6	(1.6) ^ab^	61.1	(1.4) ^a^	<0.001
Normal	66.1	(0.4) ^b^		67.1	(0.5) ^bc^	63.2	(0.7) ^a^	
Overweight	66.4	(0.5) ^b^		67.2	(0.6) ^bc^	63.5	(0.8) ^a^	
Obesity	66.8	(0.4) ^b^		67.7	(0.5) ^c^	64.2	(0.8) ^a^	
Chronic disease								
Yes	67.1	(0.3)	0.492	68.0	(0.4) ^a^	64.0	(0.5) ^b^	<0.001
No	66.7	(0.6)		67.9	(0.7) ^a^	63.2	(0.8) ^b^	
Depression								
Yes	64.7	(0.9)	0.023	65.8	(1.0) ^ab^	62.0	(1.7) ^a^	<0.001
No	66.8	(0.4)		67.7	(0.5) ^b^	63.9	(0.5) ^a^	
Perceived poor oral health								
Yes	65.0	(0.4)	<0.001	66.0	(0.5) ^a^	62.4	(0.6) ^b^	<0.001
No	67.8	(0.4)		68.6	(0.4) ^c^	65.0	(0.6) ^a^	

KHEI, Korean Healthy Eating Index. All means (SE) were calculated by applying sampling weights assigned to individual participants in the nutrition survey and health examination. ^1^ Age- and sex-adjusted means (SE). ^2^
*p*-values for mean differences of KHEI total score by factors related to KHEI using multivariate linear regression after adjusting for age (continuous) and sex. ^3^
*p*-values for mean differences of KHEI total score by area and factors related to KHEI using multivariate linear regression after adjusting for age (continuous) and sex. ^4^ Including housewives or student. Different alphabets indicate significant differences by Tukey’s test.

**Table 4 nutrients-12-01314-t004:** The regional effect on KHEI total score after adjusting for confounding factors.

Stratifying Variables	Region	β ^1^	(SE)	*p*-Value
Total	Rural (vs. Urban)	−2.6	(0.6)	<0.001
Sex				
Men	Rural (vs. Urban)	−2.5	(0.8)	0.001
Women	Rural (vs. Urban)	−2.7	(0.7)	<0.001
Age group				
65–74 yrs	Rural (vs. Urban)	−2.5	(0.7)	<0.001
≥75 yrs	Rural (vs. Urban)	−3.0	(0.8)	<0.001

KHEI, Korean Healthy Eating Index. All values were calculated by applying sampling weights assigned to individual participants in the nutrition survey and health examination. ^1^ Regression coefficient for KHEI of rural area vs. urban after adjusting for age, sex, household type, marital status, education level, household income, job status, home ownership, beneficiaries of national basic livelihood, food assistance program participation, food security, smoking, alcohol drinking, walking exercise, nutritional education, eating with others, limited social activity due to disability, weight status, chronic disease, depression, and perceived poor oral health. Sex and age were not adjusted in the analysis by sex and age group, respectively.

**Table 5 nutrients-12-01314-t005:** Factors affecting KHEI total score according to region.

	Urban (*n* = 1074)						Rural (*n* = 364)				
Risk Factors (Reference)	β ^1)^	(SE)	*p*-Value	Adj.R^2^	F-Value (*p*-Value)	Risk Factors (Reference)	β ^1)^	(SE)	*p*-Value	Adj.R^2^	F-Value (*p*-Value)
Intercept	65.4	(1.8)	<0.001	0.15	27.2 (< 0.001)		88.0	(8.2)	<0.001	0.13	12.9 (< 0.001)
Men (vs. Women)	−5.5	(1.0)	<0.001			Men (vs. Women)	−2.5	(1.1)	0.030		
Higher education level (vs. Lower level)	2.4	(0.4)	<0.001			Higher education level (vs. Lower level)	2.7	(0.7)	<0.001		
Widowed (vs. Married/living as married)	−1.4	(0.4)	<0.001			Age in years (continuous)	−0.4	(0.1)	<0.001		
Higher household income (vs. Lower income)	0.9	(0.3)	0.007			Food insecurity (vs. Food security)	5.8	(1.7)	<0.001		
Home ownership (vs. No)	3.6	(0.8)	<0.001								
Walking exercise (vs. No)	1.5	(0.7)	0.028								
Perceived poor oral health (vs. good oral health)	−2.1	(0.7)	0.002								

KHEI, Korean Healthy Eating Index. All values were calculated by applying sampling weights assigned to individual participants in the nutrition survey and health examination. In this stepwise linear regression model, age, sex, household type, marital status, education level, household income, job status, home ownership, beneficiaries of national basic livelihood, food assistance program participation, food security, smoking, alcohol drinking, walking exercise, nutritional education, eating with others, limited social activity due to disability, weight status, chronic disease, depression, and perceived poor oral health were included. *R-squared means coefficient of determination for final model.*
^1)^ Regression coefficient for KHEI for each personal variable.

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
