# Peer review of "Do Where The Elderly Live Matter? Factors Associated with Diet Quality among Korean Elderly Population Living in Urban Versus Rural Areas"

_nutrients, 2020, doi:10.3390/nu12051314_

Round 1
Reviewer 1 Report
Thank you for allowing me to revise this paper.
The Authors provided an extensive analysis of the diet quality differences and of the personal factors involved in such differences between rural and urban elderly population of Korea, based on a nationwide cross sectional survey.
My comments and suggestions are as follows:
Point 1: English language is overall correct and the results are clearly presented, just some long and complex phrases such as lines 23-25, 221-222 or 289-290 maybe should be modified, as not immediately clear
Point 2: lines 227-228, urban population appears to have a higher energetic intake from carbohydrates and fats than rural one, with the same energy intake in the two populations. Given that rural population doesn't eat more proteins and fibers than urban one, how do the Authors interpret this result?
Point 3: older rural population's fiber and protein intake appears to be lower than urban one, with higher risk of frailty and cardiovascular diseases (lines 233-234), and rural population's KHEI index is lower than urban one, but chronic diseases appear to be more common among urban population (tables 1 and 3). Which explanation do the Authors give?
Author Response
Point 1: English language is overall correct and the results are clearly presented, just some long and complex phrases such as lines 23-25, 221-222 or 289-290 maybe should be modified, as not immediately clear
Response: We modified the sentences to make them clearer. Thank you for the suggestion.
Point 2: lines 227-228, urban population appears to have a higher energetic intake from carbohydrates and fats than rural one, with the same energy intake in the two populations. Given that rural population doesn't eat more proteins and fibers than urban one, how do the Authors interpret this result?
Response: The 3rd paragraph in the Discussion described the possible reasons for this difference in protein and raw vegetable intakes. We thought that the issue of food accessibility or economic constraints in rural areas could be possible factors that explain this difference. Please let us know if you think we need to expand our discussion on this matter(Line # 236-246).
Point 3: older rural population's fiber and protein intake appears to be lower than urban one, with higher risk of frailty and cardiovascular diseases (lines 233-234), and rural population's KHEI index is lower than urban one, but chronic diseases appear to be more common among urban population (tables 1 and 3). Which explanation do the Authors give?
Response: We added one paragraph in the Discussion to address this issue (Line # 255-262).
Reviewer 2 Report
Nicely written epidemiologic paper documenting the relationship between food intake, outcome, age and urban vs rural setting in Korea.
There is no separate analysis or discussion for the very old (>85 yrs) - i wonder if this would be of value.
The finding of reduced walking activity in rural koreans is not addressed through the lens of increased manual labor; In addition the finding of reduced chronic illness is not discussed in the rural population
Author Response
Point 1: There is no separate analysis or discussion for the very old (>85 yrs) - i wonder if this would be of value.
Response: We did not perform separate analyses with the very old population in this manuscript. The reason is that the publicly available data does not provide age information who are older than 80 years old for protecting personal information. The proportion of who are older than 80 years old was 15% in the dataset used for the analyses. We think that separating this population would not give meaningful addition to our current results; however, additional data collection and analysis on this particular group might be important in future studies to examine the dietary quality and its related factors of the very old population.
Point 2: The finding of reduced walking activity in rural koreans is not addressed through the lens of increased manual labor; In addition the finding of reduced chronic illness is not discussed in the rural population
Response: We added the comments on the smaller walking activity in rural areas in the Discussion (Line #285-290). And we also commented on the possible reason for the smaller prevalence of chronic diseases in rural areas (Line #255-262).